# Transcriptomic Analysis Reveals Circadian Rhythm Homeostasis in Pearl Gentian Grouper under Acute Hypoxia

**Ren-Xie Wu [1,2]** , **Yan-Shan Liang [1]** , **Su-Fang Niu [1,2]** , **Jing Zhang [1,2,\*]** , **Bao-Gui Tang [1,2]** and **Zhen-Bang Liang [1]**

1  College of Fisheries, Guangdong Ocean University, Zhanjiang 524088, China; wurenxie@163.com (R.-X.W.); yanshan-liang@outlook.com (Y.-S.L.); wolf0487@126.com (S.-F.N.); zjtbg@163.com (B.-G.T.); liangzhenbang0403@163.com (Z.-B.L.)
2  Southern Marine Science and Engineering Guangdong Laboratory, Zhanjiang 524025, China
\*  Correspondence: zjouzj@126.com

**Abstract:** Oxygen level is an important environmental factor affecting the circadian rhythm. However, little is known about the molecular mechanism by which clock genes regulate the circadian rhythm in fish under hypoxia. To explore changes in the transcription and expression of clock genes and related molecular regulatory mechanisms in pearl gentian grouper under hypoxia, liver transcriptome data were analyzed after exposure to acute hypoxic stress (dissolved oxygen 0.5 mg/L) for 1, 3, 6, and 9 h. miR-210 and m0044-5p inhibited the expression of *period3* (*per3*) and *casein kinase 1 delta b* (*csnk1db*) in the core loop of the circadian clock, respectively. The *nuclear receptor subfamily 1 group d member 1* (*nr1d1*) and *RAR-related orphan receptor b* (*rorb*) genes in the auxiliary loop were jointly up-regulated by three miRNAs (miR-144-3p/5p, miR-361-5p, and miR-133) and the transcription factor nuclear receptor subfamily 1 group d member 2 (Nr1d2). The pearl gentian grouper maintains the stability of circadian clock systems and normal physiological metabolism under hypoxic stress by regulating the transcriptional expression of these genes via miRNAs and transcription factors to improve hypoxic tolerance. These findings provide important basic data for future research on hypoxic tolerance in pearl gentian grouper and provide new insights into the interaction between hypoxia and the circadian rhythm in fish.

**Keywords:** pearl gentian grouper; acute hypoxia; circadian rhythm; clock genes; transcriptional expression

**Key Contribution:** The pearl gentian grouper maintains physiological metabolism and enhances tolerance to hypoxia by regulating the transcriptional expression of circadian clock genes via miRNAs and transcription factors.





## 1. Introduction

The circadian rhythm is an endogenous oscillation of biological processes in organisms that follows an approximately 24 h cycle for synchronization with changes in the external environment [1,2]. Circadian rhythm regulates the physiological behaviors and metabolic reactions of diverse organisms [2]. The circadian clock is the material basis of the circadian rhythm and is composed of the input pathway, core oscillator, and output pathway [3]. The core oscillator consists of clock genes and their proteins, forming a complex core loop (also known as the transcription–translation feedback loop) [2,4]. The basic helix–loop–helix ARNT-like (Bmal)/Clock circadian regulator (Clock) heterodimer binds to the E-box element of the gene promoter, activating the clock genes and clock-controlled gene transcription (forward regulation) [5]. Subsequently, the cryptochrome circadian regulator (Cry)/Period (Per) heterodimer inhibits the transcriptional activity of the Bmal/Clock heterodimer, thereby inhibiting *cryptochrome* (*cry*) and *period* (*per*) transcription (negative regulation) [5,6]. In addition, an auxiliary feedback loop composed of the nuclear receptor

subfamily 1 group d (Nr1d) and RAR-related orphan receptor (Ror) proteins maintains the stability of the core loop [7]. The Nr1d and Ror proteins competitively bind to specific sequences of DNA, termed ROR response elements (ROREs), inhibiting and promoting the expression of the clock gene *basic helix–loop–helix ARNT-like 1* (*bmal1*), respectively [8]. Expression levels of approximately 10% to 30% of genes in animals and plants are regulated by the circadian clock system, participating in growth, development, reproduction, metabolism, and environmental adaptation [6,9]. Thus, stabilization of the circadian clock system is beneficial for the response and adaptation to periodic environmental changes [10].

Similar to other animals, fish show obvious circadian rhythms in behaviors, growth and development, and physiological metabolism [11]. Changes in environmental signals could disrupt the circadian clock system and alter the circadian rhythm in fish, leading to disruptions in behaviors and physiological processes [12,13]. Numerous studies have shown that, as the zeitgebers of the circadian clock system, light and temperature conditions could reset the clock and affect rhythms in fish [12,14]. Oxygen levels exhibit periodicity in the natural environment and affect the circadian clock system [15]. On the one hand, oxygen level fluctuations could cause changes in the amplitude of the circadian rhythm cycle and metabolism in fish [16]. Under hypoxia, the rate of hypoxia-inducible factor-1α (Hif-1α) binding to the clock gene *period1* (*per1*) increased nearly four times in zebrafish, *Danio rerio*, significantly inhibiting the expression of *per1*, leading to a change in the circadian rhythm amplitude [17]. Both hypoxia and diclofenac exposure inhibited the transcription of *clock circadian regulator* (*clock*) and *per1* and destroyed the circadian rhythm in the three-spined stickleback *Gasterosteus aculeatus*, thus affecting the transcription and activity rhythm of enzymes related to energy metabolism [18]. The metabolic rate of lake sturgeon *Acipenser fulvescens* exhibited a circadian rhythm, and its standard metabolic rate was not affected by hypoxia, but the maximum metabolic rate reduced as the oxygen level decreased [19]. These findings indicated that its metabolic rhythm was regulated by both external oxygen changes and the internal circadian rhythm [19]. On the other hand, circadian rhythm disorders would reduce the changes in physiological processes in fish under hypoxia, including changes in angiogenesis, red blood cell apoptosis, and oxygen transport capacity, ultimately leading to a decline in survival [20]. Sustained hypoxia could easily cause tissue damage in fish, while diel-cycling hypoxia could improve the ability of fish to tolerate hypoxia [21,22]. Collectively, several studies have explored the impacts of oxygen levels on the fish clock system, characterizing the complex interactions between the two. However, little is known about the molecular mechanism by which clock genes regulate the circadian rhythm in fish under hypoxia.

The pearl gentian grouper is a hybrid fish from interspecific hybridization between *Epinephelus lanceolatus* (♂) and *Epinephelus fuscoguttatus* (♀), also known as the dragon tiger grouper [23]. It has various beneficial characteristics, including strong disease resistance, a fast growth rate, high-quality meat, and rich nutrients [24]. This grouper is an economically important farmed fish in the coastal countries of Asia, particularly on the southeastern coast of China (e.g., Fujian, Guangdong, and Hainan provinces) [25]. Recently, with the promotion of high-density and intensive farming, a high stocking density, poor water exchange, and high oxygen consumption by the respiration of plankton and sediment microorganisms could easily lead to hypoxia in aquaculture waters [26]. Although this grouper has strong tolerance to hypoxia, its tolerance threshold is 0.24–0.70 mg/L [27,28]. However, hypoxia has negative effects on feeding and swimming behavior, growth and development, and physiological metabolism, and even causes a large number of deaths in fish [29]. Furthermore, hypoxia significantly alters energy metabolism, oxidative stress, and apoptosis pathways in this grouper [30,31]. As the circadian rhythm responds to environmental changes synchronously through the regulation of the circadian clock system, it plays an important role in maintaining behavioral activities, physiological metabolism, and environmental adaptation in fish [6,10]. Therefore, hypoxia would cause circadian rhythm cycle disturbances in fish, thus affecting the rhythm of their behavioral activities and physiological metabolism. In a transcriptomic analysis of the response to cold stress

in the pearl gentian grouper, we found that *nuclear receptor subfamily 1 group d member 2 (nr1d2)* and *protein phosphatase 1 catalytic subunit gamma (ppp1cc-a)* were up-regulated via ssa-miR-25-3-5p and ccr-miR-489, respectively, and contributed to the regulation of the circadian rhythm core loop to prevent cold shock and enhance cold tolerance [32]. However, the effect of hypoxia on the circadian rhythm regulation of this grouper has not been explored.

Based on the results of previous studies, we speculate that hypoxic conditions may also affect the transcriptional regulation of circadian clock genes in pearl gentian grouper, playing an important role in maintaining circadian rhythm homeostasis and normal physiological metabolism and improving its tolerance to hypoxia. This means that changes in oxygen levels are closely related to circadian rhythms. Thus, we combined previous mRNA-Seq and newly obtained miRNA-Seq data for pearl gentian grouper under acute hypoxia to obtain differentially expressed genes (DEGs), microRNAs, and transcription factors (TFs) related to circadian rhythm in this study. Using bioinformatics analyses, we further explored the transcriptional regulatory changes of the circadian clock system in the grouper under acute hypoxia. Our study provides important basic data and a reference for improving tolerance to hypoxia in this grouper under high-density and intensive farming. Furthermore, these results improve our understanding of the molecular regulatory mechanism of the circadian rhythm in fish adaptation to the hypoxic environment.

## 2. Materials and Methods

### 2.1. Acute Hypoxic Stress and Sampling

The experimental samples in this study were derived from our previous analysis of the transcriptome under hypoxic stress [30]. Briefly, 200 juvenile pearl gentian groupers (average standard length: $10.7 \pm 0.58$ cm; average standard weight: $42.8 \pm 6.99$ g) were temporarily cultured for 2 weeks at a water temperature of $25.0 \pm 1$ °C, salinity of $24.0 \pm 1$ ‰, and DO of $5.5 \pm 0.5$ mg/L. Thereafter, the water DO content in three rearing tanks was rapidly reduced from 5.5 mg/L to 0.5 mg/L within 1 h. After 0.5 mg/L DO for 1 h, 3 h, 6 h, and 9 h, three fish were randomly selected from each tank from which to take liver tissue samples; these treatment groups were named Hy1, Hy3, Hy6, and Hy9. Liver samples were taken before hypoxic stress (i.e., at 5.5 mg/L DO), named Hy0 (control group). Samples were treated with liquid nitrogen and stored at $-80$ °C until sequencing and RT-qPCR validation experiments.

### 2.2. MiRNA Sequencing and Prediction of miRNA Target Genes

High-quality RNAs from 15 liver tissues analyzed by previous transcriptome sequencing (with three duplicates per group) were sent to Genedenovo Biotechnology Co., Ltd. (Guangzhou, China) for miRNA sequencing. Differentially expressed miRNAs (DE miRNAs) were screened with $p < 0.05$ and $|\log2 (\text{Fold Change})| > 1$ as criteria. Three methods, RNAhybrid v2.1.2 + SVM_light v6.01 [33,34], miRanda v3.3a [35], and TargetScan v7.0 [36], were used to predict potential target genes of DE miRNAs. Based on our previously detected (DEGs), miRNA-mRNA regulatory pairs with negative correlations were obtained.

### 2.3. KEGG Enrichment, Pathway Network, and STEM Analyses

Based on an integration analysis of mRNA-Seq and miRNA-Seq, a KEGG enrichment analysis of genes in miRNA-mRNA regulatory pairs was performed using the OmicShare tools (https://www.omicshare.com/tools, accessed on 22 February 2023), and significant KEGG pathways related to hypoxia were selected ($p < 0.05$). Interactions among these pathways were evaluated, and hub pathways were determined by a pathway network analysis [37]. The pathway network was visualized and edited using Cytoscape v3.8.2 (Cytoscape Consortium, San Diego, CA, USA) [38].

A trend analysis of genes in miRNA-mRNA pairs was carried out using Short Time-series Expression Miner (STEM) v1.3.13 (Carnegie Mellon University, Pittsburgh, PA, USA) [39] to obtain gene expression profiles related to the duration of hypoxic stress.

The main parameters for the STEM analysis were as follows: module significance was set to $p < 0.05$, the number of modules was 20, and the minimum fold change was 2.

*2.4. Construction of a PPI Network and miRNA-TF-mRNA Regulatory Network Based on Target Genes*

Amino acid sequences of the target genes were submitted to STRING v11.5 (https://cn.string-db.org, accessed on 27 February 2023), and a protein–protein interaction (PPI) network was constructed with zebrafish as the reference species [40]. The analysis parameters were set with a confidence level of 0.9 and an FDR stringency of 0.05%.

The DEGs were aligned to the AnimalTFDB [41] using Blastp to identify potential TFs, predict target genes, and obtain TF-mRNA regulatory pairs. Based on the regulatory relationships between miRNAs, TF, and DEGs, a miRNA-TF-mRNA regulatory network was constructed using Cytoscape [38].

*2.5. RT-qPCR Validation Experiment*

Expression levels of key genes and miRNAs in 15 liver samples from one control group and four treatment groups were verified by real-time fluorescent quantitative PCR (RT-qPCR). Specific primers for genes and miRNAs were designed using Primer Premier 5 (PREMIER Biosoft International, Palo Alto, CA, USA) (Table 1). After the total RNA was extracted from each sample with TRIzol reagent (Invitrogen, Carlsbad, CA, USA), RNA quality and concentration were evaluated via 1% agarose gel electrophoresis (the extracted RNA showed in Figures S1 and S2) and NanoDrop2000 (Thermo Scientific, Waltham, MA, USA). The reverse transcriptions were performed using the One-Step gDNA Removal Kit (TransGen Biotech, Beijing, China) and miRNA First Strand cDNA Synthesis Kit (Sangon Biotech, Shanghai, China) to obtain cDNA templates for genes and miRNAs, respectively. PCR systems for genes and miRNAs were configured using the *PerfectStart*® Green qPCR SuperMix Kit (TransGen Biotech, Beijing, China) and the miRNA Fluorescence Quantitative PCR Kit (Sangon Biotech, Shanghai, China), respectively. The selected reference gene for RT-qPCR was 18S rRNA [32]. Triplicate experiments were conducted for each sample, and a negative reaction without a template was set as the control. PCR was performed using the LightCycler96 (Roche, Mannheim, Germany) under the following reaction conditions: preincubation at 95 °C for 30 s, and 40 cycles of 95 °C for 5 s and 60 °C for 30 s. The melting curve adopted instrument default values of 95 °C for 10 s, 65 °C for 60 s, and 97 °C for 1 s to determine primer dimer and non-specific amplification. Relative expression levels of genes and miRNAs were calculated using the comparative CT method ($2^{-\Delta\Delta CT}$), and line charts were drawn using GraphPad Prism v8.0.2 (GraphPad Software, San Diego, CA, USA).

**Table 1.** Primer sequences of key genes and miRNAs.

| Genes/miRNAs | Forward Primer (5′-3′) | Reverse Primer (5′-3′) |
|---|---|---|
| 18S rRNA | GACTCTGGCGTGCTAACTA | CAATCTCGTGTGGCTGAAC |
| *period3 (per3)* | AAGTGGGGCAGGAAGATGAA | TTCTTCATCTCAGCCACCGT |
| *casein kinase 1, delta b (csnk1db)* | ACCACGAGGAACCCAAGACG | CACGCTCCATTCCAGACACC |
| *RAR-related orphan receptor b (rorb)* | ACACACTCCTGGGACTTCTG | CGAACTAACCGTAACCGCTG |
| *nuclear receptor subfamily 1, group d, member 1 (nr1d1)* | AGTGCATGTGTGTCAGAGGT | CACATTCACCCGCTCATCAG |
| *nuclear receptor subfamily 1, group d, member 2 (nr1d2)* | AGTGCATGTGTGTCAGAGGT | CACATTCACCCGCTCATCAG |
| miR-361-5p | TTATCAGAATCTCCAGGGGTCC | |
| miR-210 | CTGTGCGTGTGACATCGGCT | |
| m0044-5p | GTGTTCAGTCTGTTGGTCCGTCT | |
| Universal primer R | | GTGCAGGGTCCGAGGT |

## 3. Results

*3.1. KEGG Enrichment and Pathway Network Analyses*

After data filtering, mRNA and miRNA sequencing yielded clean reads ranging from 35,152,534 to 66,514,474 and 8,691,172 to 26,262,181, respectively. In total, 2383 DEGs and

91 DE miRNAs were screened from mRNA and miRNA sequencing data, respectively. By combining the mRNA-Seq and miRNA-Seq datasets, a total of 562 miRNA-mRNA regulatory pairs were obtained, including 72 DE miRNAs and 358 DEGs. Based on a KEGG enrichment analysis of 358 DEGs, the circadian rhythm pathway was the most significant pathway except for cancer-related pathways among the top 20 pathways (Figure 1). During the whole hypoxic stress process, we did not observe disease symptoms in the experimental fish. The four genes *period3* (*per3*), *casein kinase 1, delta b* (*csnk1db*), *nuclear receptor subfamily 1, group d, member 1* (*nr1d1*), and *RAR-related orphan receptor b* (*rorb*) were significantly enriched in the circadian rhythm pathway, and these were likely to play an important role in the acute hypoxic response in pearl gentian grouper. Further pathway network analysis revealed that the circadian rhythm pathway (ko04710) was connected with other pathways by directly acting on ubiquitin-mediated proteolysis (ko04120) and the MAPK signaling pathway (ko04010), thereby regulating the acute hypoxia response in pearl gentian grouper (Figure 2). Moreover, ko04710 was associated with other circadian-related pathways, including circadian rhythm-fly (ko04711), circadian entrainment (ko04713), and African trypanosomiasis (ko05143), although these three pathways were not connected to other pathways.

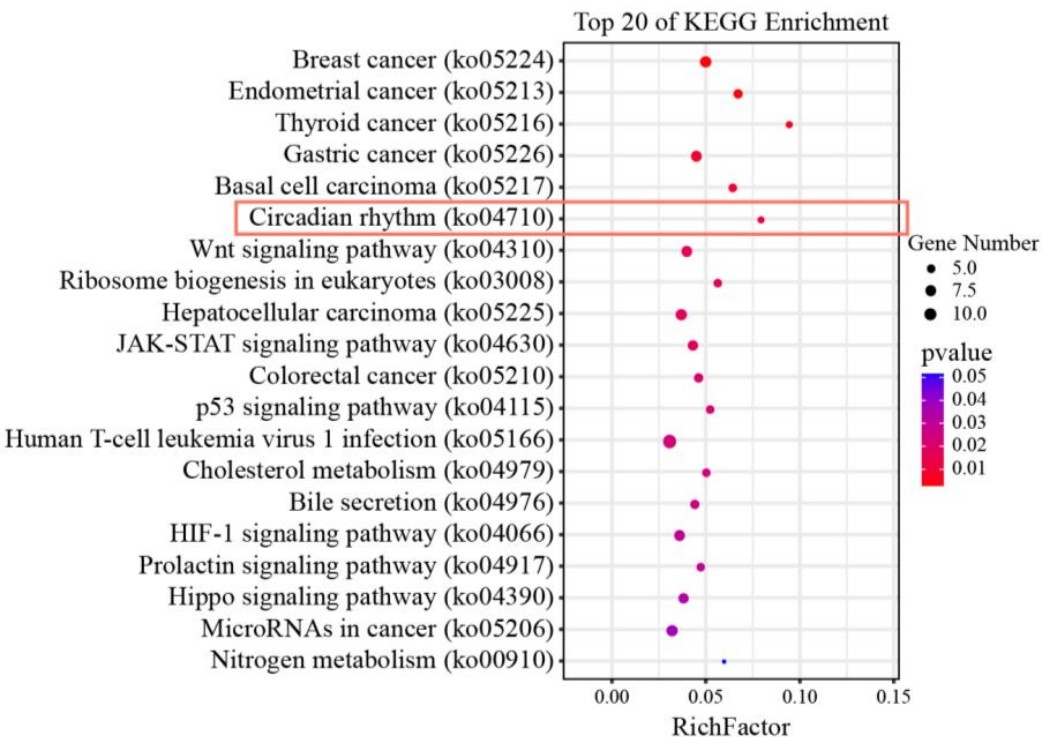

**Figure 1.** KEGG enrichment analysis of 358 target genes.

### 3.2. Expression Trends of Target Genes

In the STEM analysis, 358 target genes (from 562 miRNA–mRNA regulatory pairs in the integration analysis) were clustered into 20 gene expression patterns (Figure 3). Three expression profiles were significant ($p < 0.05$) and contained 63.41% of the target genes. The expression levels of 122 genes in Profile 0 decreased consistently during acute hypoxic stress, whereas the expression levels of 31 genes in Profile 19 continued to increase. In Profile 18, the expression levels of 74 genes increased in the early stage of hypoxia (0–3 h) and then decreased slightly (6–9 h), with an overall increasing trend. The expression patterns of the four circadian rhythm genes belonged to Profile 0 (*per3* and *csnk1db*), Profile 19 (*nr1d1*), and Profile 17 (*rorb*). Obviously, the diversity of gene expression patterns indicates that a complex regulatory mechanism was involved in the response to hypoxia in pearl gentian grouper.

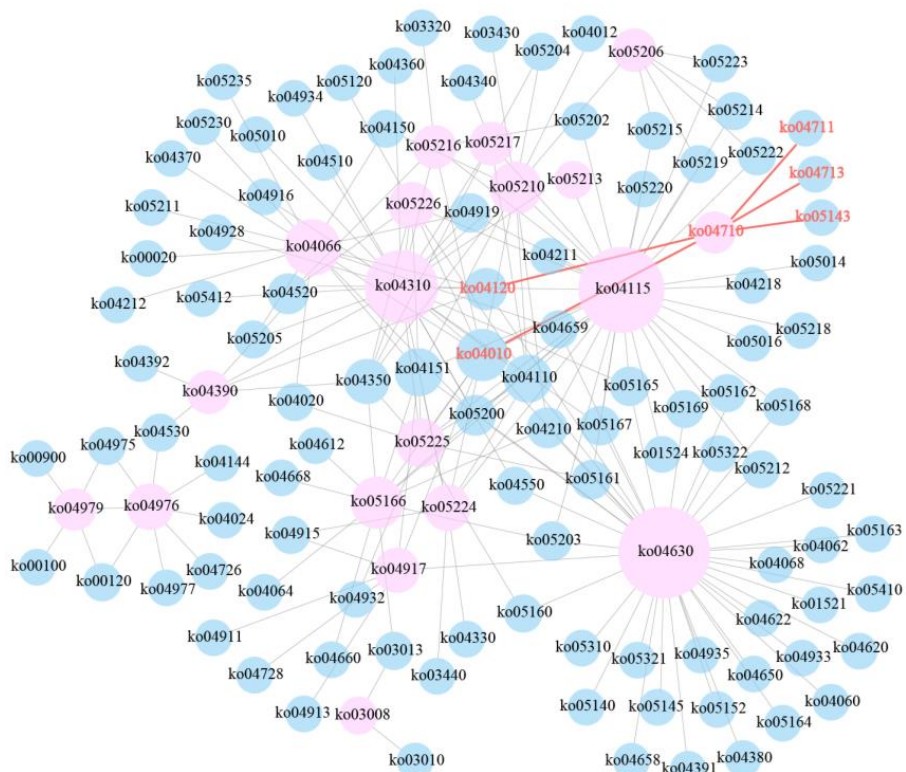

**Figure 2.** Pathway interaction network. The nodes represent pathways (pink represent significantly enriched pathways, blue represent non-significant pathways), the size of each node represents its connectivity (where a larger node indicates a more important pathway in the network), and the lines represent the interactions between pathways.

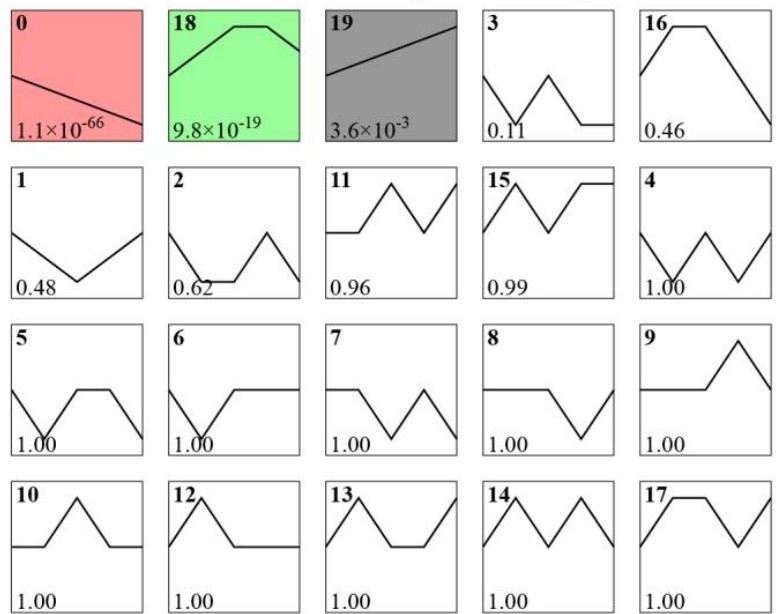

**Figure 3.** Twenty gene expression profiles involving 358 target genes. The number at the bottom left represent *p*-value, colors represent significant profiles (*p* < 0.05), white represent non-significant profiles.

### 3.3. PPI Network and miRNA-TF-mRNA Regulatory Network of Circadian Rhythm Genes

A PPI network analysis of four circadian rhythm genes (*per3*, *csnk1db*, *nr1d1*, and *rorb*) was performed using zebrafish as the reference species. As shown in Figure 4, the PPI network was composed of 24 proteins with direct or indirect interactions, mainly related to the circadian rhythm. Among these, 4, 1, and 13 proteins showed direct interactions with Period3 (Per3), Casein kinase 1 delta b (Csnk1db), and Nuclear receptor subfamily 1 group d member 1 (Nr1d1), respectively. However, there were no interactions between RAR-related orphan receptor b (Rorb) and other proteins in the PPI network. This may be explained by the limited research on the functional role of Rorb in fish.

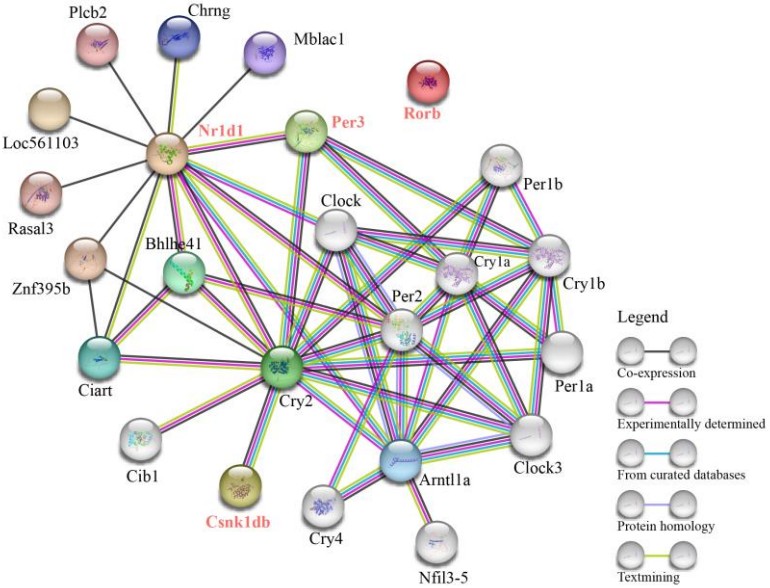

**Figure 4.** Protein–protein interaction network of the four circadian rhythm genes with zebrafish as the reference species.

The TFs and their target genes formed 33 TF-mRNA regulatory pairs, including 28 TF and 24 DEGs. Combined with 562 miRNA-mRNA pairs, we detected seven regulatory pairs related to the circadian clock (Figure 5). The miRNA-TF-mRNA regulatory network included six DE miRNAs, one TF, and four DEGs. The *per3* and *csnk1db* genes were regulated by miR-210 and m0044-5p, respectively. The *nr1d1* gene was co-regulated by miR-144-3p, miR-144-5p, and miR-361-5p. *rorb* was co-regulated by miR-133 and the TF Nuclear receptor subfamily 1group d member 2 (Nr1d2).

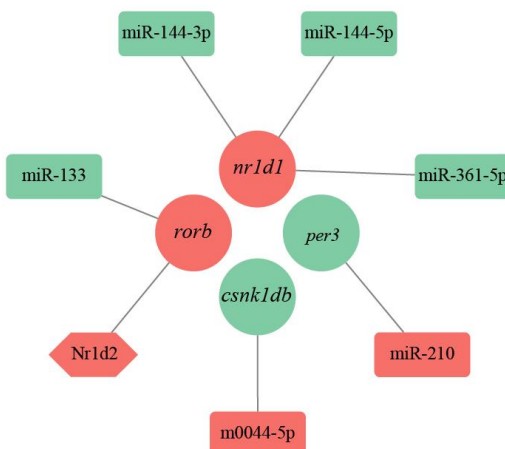

**Figure 5.** miRNA-TF-mRNA regulatory network for the circadian rhythm.

### 3.4. RT-qPCR Validation of Key Genes and miRNAs

Relative expression levels of four circadian rhythm genes (*per3*, *csnk1db*, *nr1d1*, and *rorb*) and four transcriptional regulators (miR-210, m0044-5p, miR-361-5p, and TF Nr1d2) in all groups were quantitatively verified by RT-qPCR. As illustrated in Figure 6, the expression levels of *nr1d1*, *rorb*, miR-210, and m0044-5p were up-regulated, while the expression levels of *per3*, *csnk1db*, and miR-361-5p were down-regulated overall. The *nr1d2* expression fluctuated. RT-qPCR expression levels of genes and miRNAs were consistent with the relative expression trends obtained by sequencing, supporting the reliability and accuracy of mRNA-Seq and miRNA-Seq results.

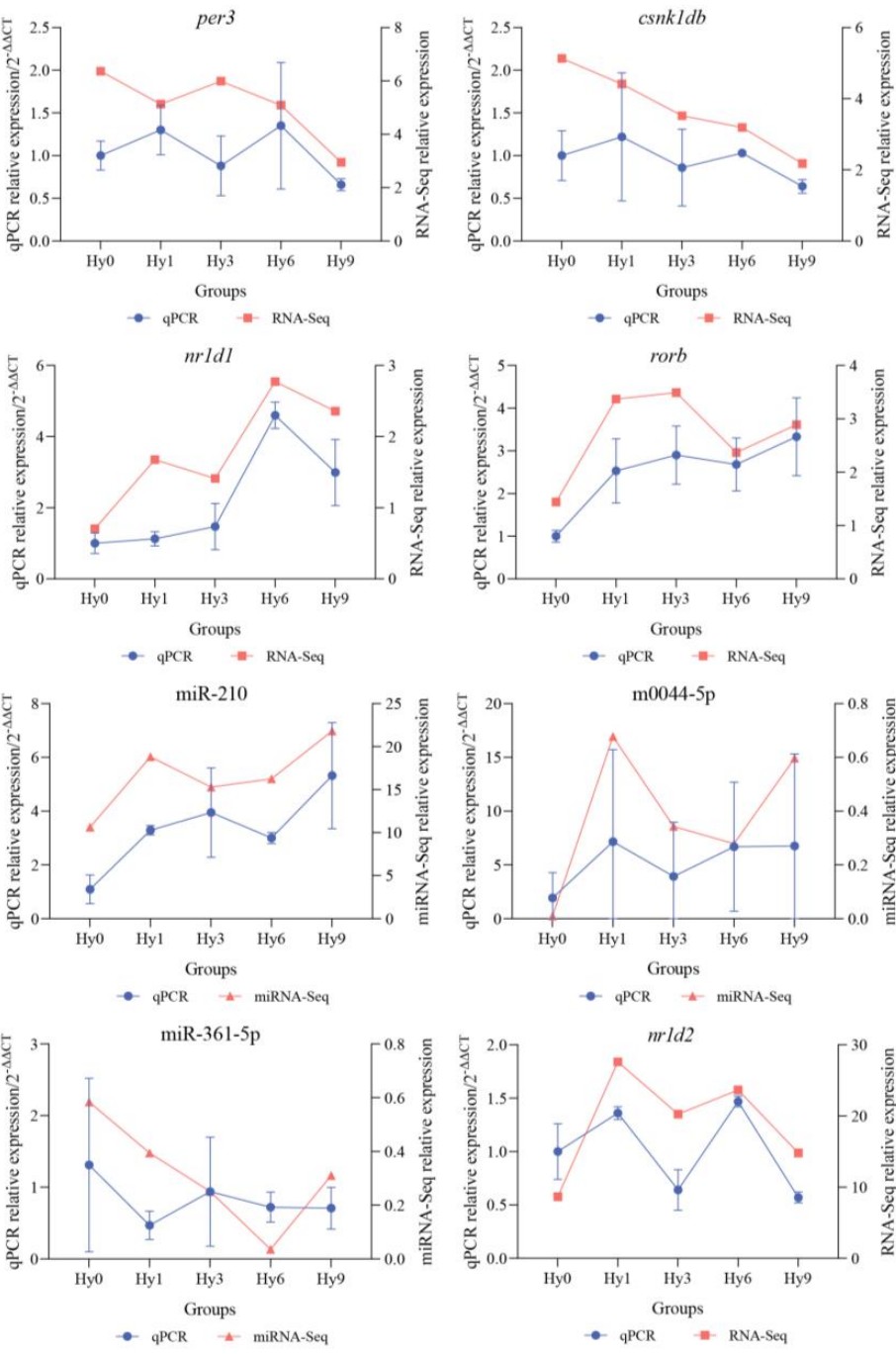

**Figure 6.** Comparison of relative expression levels obtained by RT-qPCR validation and sequencing.

## 4. Discussion

Among the 358 DEGs detected in this study, the four genes *per3*, *csnk1db*, *rorb*, and *nr1d1* were significantly enriched in the circadian rhythm pathway. Moreover, regulatory relationships between these four target genes and seven regulatory factors (miR-210, m0044-5p, miR-133, miR-144-3p, miR-144-5p, miR-361-5p, and TF Nr1d2) were determined. By further exploring the functions and pathways related to these molecules, we confirmed that they play crucial roles in regulating the circadian rhythm cycle and circadian clock system of pearl gentian grouper exposed to acute hypoxia.

*4.1. Expression Changes of Core Clock Genes and Stability of the Circadian Cycle under Hypoxia*

*Period* genes (including *per1*, *per2*, and *per3* isoforms) are negative regulatory elements of the circadian clock [42,43]. Zheng et al. [44] demonstrated that the double knockout of the *per1* and *per2* genes leads to the loss of circadian rhythm in mice. That is, these two genes play a direct and important role in the regulation of the animal circadian rhythm [45]. The precise regulatory roles of *per1* and *per2* in the circadian rhythm of fish under hypoxia and cold stress have been confirmed by several studies [17,20,46]. However, the functional role of *per3* in the animal circadian rhythm is still controversial. Shearman et al. [47] found that a *per3* deficiency does not directly cause the loss of the circadian rhythm in mice, suggesting that the gene is not necessary for the circadian rhythm. Furthermore, Bae et al. [48] reported that the changes in the circadian rhythm in mice with *per1/per3* and *per2/per3* double deficiencies were similar to those of mice with *per1* or *per2* deficiency alone. In contrast, Yagita et al. [49] confirmed that the Per3 protein encoded by *per3* promotes nuclear translocation and stability of Per1 and Per2 in mammals. In a circadian rhythm transcriptome study of the gilthead sea bream *Sparus aurata*, Yúfera et al. [50] found that the expression peaks of *per3* appeared before those of *per1* and *per2*, indicating that *per3* functions in the regulation of *per1* and *per2*. Recently, *per3* expression in various tissues of the darkbarbel catfish *Pelteobagrus vachellii* was observed to show obvious circadian rhythms [51]. In the liver of pearl gentian grouper under hypoxia, the up-regulation of miR-210 inhibited the expression of *per3* (Figure 7). MiR-210, which is a hypoxia-induced miRNA with an essential regulatory role in the animal cell cycle, DNA damage repair, angiogenesis, stem cell differentiation, mitochondrial metabolism, and cancer treatment [52]. It is important for cells to maintain normal metabolic function in hypoxic environments [52]. Similarly, Nagel et al. [53] reported that miR-192 and miR-194 could inhibit the transcriptional expression of the *period* gene family, which may shorten the circadian rhythm period in animals. Hung et al. [46] found that cold tolerance in zebrafish larvae could be improved by regulating the expression of *per2* via dre-miR-29b. Therefore, we infer that hypoxia induces the up-regulation of miR-210 expression in pearl gentian grouper to inhibit the expression of *per3*, resulting in decreased period protein levels and Cry/Per heterodimer activity, thereby weakening the inhibitory effect on the Bmal/Clock heterodimer [2,54]. This change ultimately shortens the length of its circadian rhythm cycle, which is conducive to enhancing tolerance to hypoxia in pearl gentian grouper. To our knowledge, this study reveals, for the first time, the functional role of *per3* in enhancing hypoxia tolerance in fish by regulating the circadian rhythm.

We also observed that the up-regulation of the novel miRNA m0044-5p inhibited the expression of the *csnk1db* gene (Figure 7). The CK1δ kinase encoded by *csnk1db* belongs to the casein kinase family, which directly promotes the degradation of PER or changes its location in cells via phosphorylation [55,56]. Phosphorylation modification is an important regulatory mechanism for the circadian clock, affecting the stability, activity, and interactions of circadian clock proteins [57]. When the activity of CK1δ kinase is inhibited or deactivated, the animal circadian cycle is prolonged [58]. Its overexpression would shorten the circadian cycle [59]. Moreover, CK1 kinase cooperates with protein phosphatase 1 (PP1) to jointly regulate the activity and stability of a variety of circadian clock proteins and participate in regulating the circadian cycle [60,61]. Smadja Storz et al. [62] found that the rhythmic expression of *per* and *arylalkylamine N-acetyltransferase 2* (*aanat2*) in ze-

brafish disappears with the inhibition of CK1δ kinase activity, with effects on the behavioral rhythm. Miao et al. [32] reported that ccr-miR-489 in the liver of pearl gentian grouper shortens the circadian cycle by regulating the transcriptional expression of *ppp1cc-a* and participating in the phosphorylation of PER under cold stress. The *csnk1db* gene regulates the activity and stability of circadian clock elements and determines the length of the circadian cycle by the autophosphorylation of CK1 kinase or in collaboration with other genes [63]. Therefore, the down-regulation of *csnk1db* means that the phosphorylation of the CK1 kinase was inhibited by hypoxia in pearl gentian grouper, thereby reducing the phosphorylation and degradation of the Cry/Per heterodimer. This could lead to increased Bmal/Clock heterodimer activity [58,64], thus prolonging the length of the circadian cycle in this grouper.

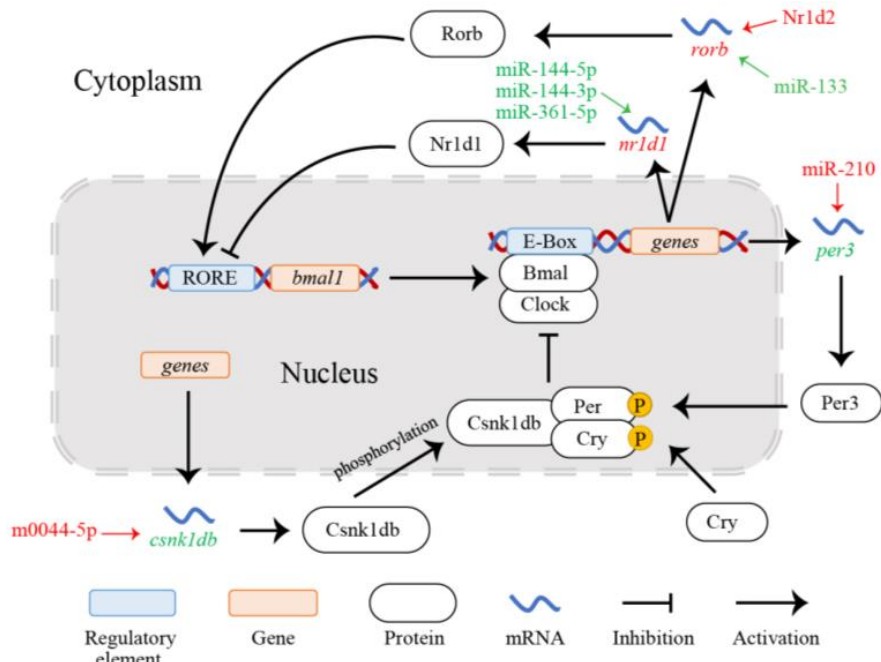

**Figure 7.** Regulatory network for the circadian rhythm in pearl gentian grouper under hypoxia.

To sum up, miR-210 and m0044-5p inhibited the transcription of *per3* and *csnk1db*, respectively, and shortened and extended the circadian cycle, maintaining the stability of the circadian cycle of pearl gentian grouper under hypoxia, which is conducive to physiological and biochemical reactions and survival. This regulatory relationship between oxygen and the circadian clock is similar to the temperature compensation mechanism of the circadian rhythm [65]. That is, a change in the oxygen content does not have a significant impact on the length of the circadian cycle in animals within a certain level and duration of hypoxia. Circadian clock genes are capable of self-regulation to maintain the stability of the circadian cycle under environmental change.

### 4.2. Regulation of the Auxiliary Loop and Stability of Circadian Clock Systems

The auxiliary loop of the circadian clock is mainly mediated by the Nr1d and Ror proteins, which regulate the transcriptional activity of the clock gene *bmal1* in the core loop by competitive binding [66]. In this study, we found that the down-regulation of three miRNAs (i.e., miR-144-3p, miR-144-5p, and miR-361-5p) collectively promotes the expression of the *nr1d1* gene (Figure 7). The *nr1d1* gene is a regulatory gene in the circadian clock system; it is expressed in various tissues and exhibits periodic changes [67]. The Nr1d1 protein encoded by *nr1d1* inhibits *bmal1* gene expression by binding to the promoter RORE element, thus participating in the circadian rhythm and lipid metabolism to maintain the normal metabolic function of animals [8,68]. Previous studies have shown that *nr1d1*

deficiency could disrupt the circadian rhythm, affect the expression of core clock genes and downstream genes, and influence glucose and lipid metabolism in animals [68,69]. The knockout of *nr1d1* altered the expression of circadian rhythm genes and metabolism in the liver of mice [69]. The *nr1d1* gene also plays a pivotal role in the circadian clock system and movement rhythm in various fishes, such as zebrafish [70], Atlantic cod *Gadus morhua* [71], Chinese perch *Siniperca chuatsi* [72], and the goldfish *Carassius auratus* [66]. Three regulators (miR-144-3p, miR-144-5p, and miR-361-5p) of the *nr1d1* gene participate in the regulation of the circadian rhythm via clock genes or related signaling pathways [73,74]. Therefore, the above three miRNAs co-regulated *nr1d1*, thereby inhibiting the transcriptional expression of *bmal1* and facilitating circadian rhythm stability and normal physiological metabolism under hypoxic conditions in pearl gentian grouper. In the previous cold stress transcriptome analysis, we detected increased *nr1d2* gene expression via ssa-miR-25-3-5p, and this gene participated in the maintenance of the circadian rhythm balance and lipid homeostasis to enhance cold tolerance [32]. As a homologue of *nr1d1*, *nr1d2* is also considered to be involved in the regulation of the circadian rhythm, energy metabolism, cellular autophagy, and other biological processes [68,69]. The *nr1d2* gene was not significantly enriched in the circadian rhythm pathway, in our study. However, the results of the previous cold stress analysis suggest that *nr1d1* and *nr1d2* may both contribute to the regulation of circadian rhythms under stress and may have complementary roles [32].

Furthermore, we observed that the up-regulation of *rorb* in the auxiliary loop is jointly regulated by the TF Nr1d2 and miR-133 (Figure 7). The *rorb* gene is a member of the *ror* gene family and encodes the retinoid-related orphan receptor involved in the auxiliary loop and maintaining the stability of the core loop by binding to the RORE element of *bmal1* and activating gene expression [8,75]. The RORE element of *rorb* binds to the Nr1d protein, and its transcriptional expression is regulated by Nr1d [76]. Masana et al. [77] reported that mice lacking Rorb showed abnormal circadian behavior, confirming that *rorb* plays a critical role in animal circadian regulation. Thus, we believe that the TF Nr1d2 up-regulated the expression of *rorb*, which is beneficial for the stable regulation of the circadian clock system in the pearl gentian grouper under hypoxia. In addition, miR-133, which regulates *rorb*, is involved in the regulation of embryonic and vascular development, muscle regeneration, cardiac rhythm, and energy metabolism in fish [78]. It has been demonstrated that miR-133-3p participates in the regulation of glucose and lipid metabolism in the largemouth bass *Micropterus salmoides* during acute hypoxia [79]. Although the function of miR-133 in the circadian clock system has not been established, our results show that *rorb* is a potential target gene for miR-133, and the down-regulation of miR-133 significantly increases the expression of *rorb*. This suggests that miR-133 plays a key role in the regulation of the circadian rhythm in pearl gentian grouper under hypoxia. Based on the above analysis, we conclude that *rorb* activates the transcriptional expression of *bmal1* under the co-regulation of the TF Nr1d2 and miR-133 to maintain the normal circadian rhythm of pearl gentian grouper exposed to acute hypoxia.

In summary, various miRNAs, TFs, and target genes mediate the transcriptional expression of the core gene *bmal1* to stabilize the core loop of the circadian clock, thereby preventing disruptions of the circadian rhythm caused by hypoxia in the pearl gentian grouper. This transcriptional regulation would be conducive to the survival of this grouper under hypoxia and the maintenance of normal physiological and metabolic activities.

## 5. Conclusions

Based on analyses of mRNA and miRNA data, we detected a close correlation between changes in the circadian rhythm of pearl gentian grouper and acute hypoxia and obtained insight into the molecular regulatory mechanism underlying the circadian rhythm under hypoxia. Clock-related genes (*per3*, *csnk1db*, *nr1d1*, and *rorb*) maintained the circadian rhythm cycle and the overall stability of the circadian clock system via multiple miRNAs and TF Nr1d2. This self-regulation of the circadian rhythm by the grouper with a certain intensity and duration of hypoxia contributes to the maintenance of normal physiolog-

ical metabolism and enhanced hypoxia tolerance. These findings provide an important reference for improving hypoxic tolerance in pearl gentian grouper under high-density and intensive farming. Notably, the role of the *per3* gene and miR-133 in the regulation of the circadian rhythm in the fish response to hypoxia was identified for the first time in this study. Collectively, these results improve our understanding of the regulation of fish circadian rhythm under environmental stress.

**Supplementary Materials:** The following supporting information can be downloaded at: https://www.mdpi.com/article/10.3390/fishes8070358/s1, Figure S1: RNA extractions and migration-1; Figure S2: RNA extractions and migration-2.

**Author Contributions:** Conceptualization, R.-X.W., J.Z. and S.-F.N.; methodology, R.-X.W., S.-F.N. and Y.-S.L.; validation, R.-X.W. and Y.-S.L.; formal analysis, Y.-S.L.; investigation, Y.-S.L., J.Z. and Z.-B.L.; data curation, R.-X.W. and Y.-S.L.; writing—original draft preparation, R.-X.W. and Y.-S.L.; writing—review and editing, R.-X.W., J.Z. and Y.-S.L.; visualization, R.-X.W. and Y.-S.L.; supervision, R.-X.W., J.Z., S.-F.N. and B.-G.T.; project administration, R.-X.W., J.Z. and B.-G.T.; funding acquisition, J.Z., R.-X.W. and B.-G.T. All authors have read and agreed to the published version of the manuscript.

**Funding:** This research was funded by the Fund of Southern Marine Science and Engineering Guangdong Laboratory (Zhanjiang), China (No. ZJW-2019-06); the Science and Technology Planning Project of Guangdong Province, China (No. 2017A030303077).

**Institutional Review Board Statement:** The animal study protocol was approved by the Institutional Review Board (Ethics Committee) of the Animal Experimental Ethics Committee of Guangdong Ocean University (approval number: 0301-2020).

**Data Availability Statement:** mRNA-Seq and miRNA-Seq raw reads data have been uploaded to NCBI database (accession: PRJNA801908 and PRJNA967800).

**Conflicts of Interest:** The authors declare no conflict of interest.

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
