# Peer review of "Transcriptomic Analysis Reveals Circadian Rhythm Homeostasis in Pearl Gentian Grouper under Acute Hypoxia"

_fishes, doi:10.3390/fishes8070358_

Round 1
Reviewer 1 Report
See in attached file.

I didn't notice any major errors in the writing in English.
I seem to have seen a few typos (not indicated in my document), and proofreading by the authors as part of the revisions should enable them to be corrected in the final rendering.
Reviewer 2 Report
Dear authors,
This manuscript is similar in nature and indeed is using the same samples and results already obtained for the publications available at https://doi.org/10.1016/j.aquaculture.2022.738635. I recommend you to clarify better this issue, otherwise, it is like you are adding an additional piece of information to something you could already have done in the previous paper.
The main issues I found in your manuscript are the following:
- When writing down gene and protein names from fish you should follow zebrafish nomenclature guidelines available at https://zfin.atlassian.net/wiki/spaces/general/pages/1818394635/ZFIN+Zebrafish+Nomenclature+Conventions. In summary, the fish genes should be written in small case letters, in italics, without Greek letters, or Latin numerals. The acronyms should also follow these rules, but additionally, no hyphens should be used. Proteins and their acronyms are written without italicization and the first letter should be capitalized. Moreover, the first time use of an acronym should be preceded by the name of the gene of the protein in full. Be sure the names of the genes and proteins follow zebrafish names as in https://zfin.org/
Examples of this are:
Line 15: period 3 (per3) and casein kinase 1, delta ( csnk1d)
Line 16: The nuclear receptor subfamily 1, group d, member 1 (nr1d1) and RAR-related orphan receptor B (rorb)
Line 18: nr1d1
Line 36: The Basic helix-loop-helix ARNT like (BMAL)/Clock circadian regulator (CLOCK) heterodimer... (for human genes and protein the acronyms are ok, but no description is given before the use of the acronym)
Line 38: the Cryptochrome circadian regulator (CRY)/ Period (PER) heterodimer...
Lines 43-44: (they correspond to reference 8 which is from zebrafish) The Nuclear receptor subfamily 1, group d (Nr1d) and RAR-related orphan receptor (Ror)
I hope these examples can enlighten you on how to properly write the names of the genes and proteins when talking about mammals and fish.
Others:
RORE elements? define them
Line 77: Pearl gentian grouper is not a new variety, but the hybrid or is produced by hybridization. Change the sentence accordingly
Reference 24 in line 80 is not the best reference because is not on the beneficial characteristics, including its strong disease resistance, ... Use another more appropriate reference
Reference 29 in line 89 is not the best reference. The reference is a whole book, and that makes no sense
I detected a lot of self-citations in this manuscript, like in references 31, 33, etc.
The goal of this research should be better explained. What is written in lines 105-107 is not enough
In the material and methods sections, there are parts summarized from reference 31. Remove all the non-essential information, given you are calling that reference
Line 125: it should read -80 °C. Between number and unit, you should always add a space. See page 149 from https://www.bipm.org/utils/common/pdf/si-brochure/SI-Brochure-9.pdf (Formatting the value of a quantity). This includes °C
What is the difference between subparagraph 2.3 in this manuscript and 2.5 in reference 31? I do not see much. Do not repeat.
Lines 149-152: a PPI network was constructed using zebrafish as the reference species. The grouper is a diploid species, but zebrafish suffered an additional genome duplication. How can this affect the obtained results? Was this taken into account?
Subparagraph 2.5: Why those specific genes were chosen? why not clock, bmal, or cry? There is a lot of information missing in this section of the manuscript. Include all relevant information from MIQE guidelines available at doi: 10.1373/clinchem.2008.112797. Why rna18s was used as the internal reference gene and not actb as in reference 31? In order to use a gene as an internal reference you should demonstrate its stability during the experiment. How you did it? (It is not valid that the gene has been used by other authors, given the validations have to be done in each experimental condition). It is part of the MIQE guidelines.
Table1: Follow zebrafish nomenclature guidelines
In Figures 1 and 2 it is not clear why did you highlight those pathways (ko04710, ko04120, and ko04010) and not the others? I can see that ko04710 is also connected to ko04711, 04713, and 05143. Why leave those outside?
Figure 3: is very similar to Figure 6 from reference 31. Explain, because it is not clear in the figure or the text
Subparagraph 3.4: Make a correlation analysis between both types of approaches. Just to say that relative expression levels were verified is not enough. How?
Figure 6: This is not a time-course experiment, then I do not understand why it has been chosen this type of representation. It should be represented better in bars.
I believe the conclusions are weak for the amount of work involved. just to say that 3 microRNAs, 1 TF, and 4 circadian rhythm genes are involved. Even you did not study the real protein interactions here
Only minor issues in the English language have been detected in the manuscript. In general, the manuscript is well written, but it does not follow the fish gene and protein nomenclature guidelines, which they should follow the ones for zebrafish.
In the introduction, there is a sentence that from my point of view is not correct: "The circadian rhythm is a process in which changes in physiological behaviors and metabolic reactions follow an approximately 24 h cycle for synchronization with changes in the external environment." and it should be changed to a more comprehensive sentence
Round 2
Reviewer 2 Report
NONE